# Different Effects of Reactive Species Generated from Chemical Donors on Seed Germination, Growth, and Chemical Contents of *Oryza sativa* L.

**DOI:** 10.3390/plants12040765

**Published:** 2023-02-08

**Authors:** Thanyarat Chuesaard, Penpilai Peankid, Suwannee Thaworn, Anuwat Jaradrattanapaiboon, Mayura Veerana, Kamonporn Panngom

**Affiliations:** 1Basic Science, Maejo University Phrae Campus, Rong Kwang, Phrae 54140, Thailand; 2Forest Management Program, Maejo University Phrae Campus, Rong Kwang, Phrae 54140, Thailand; 3Agroforestry Program, Maejo University Phrae Campus, Rong Kwang, Phrae 54140, Thailand; 4Crop Production Technology Program, Maejo University Phrae Campus, Rong Kwang, Phrae 54140, Thailand; 5Department of Applied Radiation and Isotope, Faculty of Science, Kasetsart University, Bangkok 10900, Thailand

**Keywords:** hydrogen peroxide, sodium nitroprusside, sodium nitrite, reactive oxygen and nitrogen species, seed germination and growth, *Oryza sativa* L.

## Abstract

Reactive oxygen and nitrogen species (RONS) play an important role as signaling molecules in redox reactions throughout a plant life cycle. The purpose of this study was to assess how hydrogen peroxide (H_2_O_2_), a reactive oxygen species (ROS) and reactive nitrogen species (RNS) generated from sodium nitroprusside (SNP) and sodium nitrite, affects the germination, growth, and chemical contents of two rice cultivars (Pathum Tani and Sanpatong). The results showed that RNS generated from chemical donors and, especially, H_2_O_2_, enhanced the germination of the studied rice cultivars. Among the three chemical donors, H_2_O_2_ showed the best efficacy of the reactive species for activating early seed germination, followed by sodium nitrite and SNP. The highest percentage of seed germination rose to 99% at 6 h germination time after treatment with 25 mM of H_2_O_2_ for 24 h. Moreover, H_2_O_2_ produced a significant increase in the α-amylase activity and total soluble proteins. It was observed that a treatment with H_2_O_2_ on germinated seeds produced radicles with a dark blue color for longer than treatments with sodium nitrite and SNP. Our findings imply that H_2_O_2_ had a critical role in improving the germination and altering the chemical contents of rice seeds.

## 1. Introduction

The germination and dormancy processes of seeds are important steps for long-term seed survival [1]. Seed germination is the mechanism of initial growth in a plant’s life; it is also a complex physiological event for a seed. Likewise, seed dormancy is the pathway of the resting seed and occurs as a response to an unsuitable environment [2]. These processes have complicated metabolisms and are related to extrinsic and intrinsic factors in the environment and genetic materials, such as moisture, oxygen, light, carbon dioxide, phytohormones, genes, seed structure, etc. [3,4]. Abcisic acid (ABA) and Gibberellic acids (GAs) are the major phytohormones that are related to the germination and dormancy processes of plants [1]. Generally, ABA determines seed dormancy and inhibits seeds from germination, while GAs are necessary for seed germination enhancement, which is dependent on their ratio [2]. Moreover, it is well known that RONS are produced as by-products from the metabolism in plants and have an important role in endosperm weakening, the mobilization of seed reserves, protection against pathogens, the defense against biotic and abiotic stress, and programmed cell death [4,5]. However, RONS perform oxidative and nitrosative signaling to induce seed germination within the oxidative pathway level and can play both beneficial and harmful functions depending on the dose and the plant species [4].

During the seed germination stage, RONS are essential for stimulating the physiological aspects of the seed and the developmental processes of the seedling [6], whereas a high level of reactive species triggers senescence, cell death, cell cycle arrest, and other harmful reactions [7]. Exogenous reactive species such as superoxide radicals (O_2_^•−^), hydroxyl radicals (^•^OH), hydrogen peroxide (H_2_O_2_), peroxyl radicals (ROO^•^), nitric oxide (NO), and peroxynitrite (OONO^−^) are generated from ROS and RNS chemical donors. Those molecules are called free radicals, which are molecules containing an unpaired electron in their external molecular orbital. They are reactive and short-lived intermediates [8]. Many recent studies have mentioned that chemically generated and exogenously applied reactive species can play critical roles during seed germination, growth, and changes in the seed metabolism, as displayed in many plants [9,10,11]. The role of reactive species is not only ameliorating germination, growth, and metabolism in plants, but also in influencing all phases of the seed life cycle [12]. The various metabolisms inside the seed during germination are activated to speed up germination by RONS, which are generated from inside and outside the seeds [11].

These studies suggest that exogenous reactive oxygen and nitrogen species generated from a chemical donor are able to induce and stimulate early seed germination, subsequent seedling growth, and the seed metabolism in rice seeds. However, there are still a limited number of studies on the comparative analysis of individual reactive species generated from different chemical donors and their effects on different rice cultivars. Therefore, this research focuses on reactive oxygen and reactive nitrogen species to determine what types of reactive species as well as their doses serve as activators for seed germination and growth in non-glutinous (Pathum Tani) and glutinous (Sanpatong) rice. Additionally, we analyzed the chemical contents and the accumulation of ROS during seed germination.

## 2. Results

### 2.1. Reactive Oxygen and Nitrogen Species Differently Enhanced Seed Germination and Growth

Rice seeds (Pathum Tani and Sanpatong) were treated with reactive species generated from different chemical donors in several concentrations for the activation of early seed germination. Figure 1A shows the germination percentage for the rice seed cultivar Pathum Tani after a treatment with H_2_O_2_ solutions for 12 and 24 h. The results showed that the percentage of seed germination was significantly increased (*p* ≤ 0.01), especially at a 25 mM concentration. Rice seeds being treated with H_2_O_2_ for 24 h accelerated the speed of the germination percentage at 3 h of germination time (94%) and the highest (99%) at 6 h of germination time. Moreover, a long root was found after seed treatment with a H_2_O_2_ solution. When the concentration of H_2_O_2_ increased, the germination percentage decreased, but it was still higher than the control. Figure 1B shows the percentage of seed germination after receiving treatment for 12 and 24 h with SNP solutions. The percentages of seed germination were not different among the several SNP concentrations after either of the treatment times, except for treatment with 12.5 µM for 24 h, which reached approximately 52.22% at 6 h of germination time. Figure 1C shows the percentage of seed germination after treatment with sodium nitrite solutions for 12 and 24 h. The percentage of seed germination was not different from the control for the 12 h treatment time. However, the percentage of seed germination dramatically increased up to 100% when rice seeds were treated with sodium nitrite solutions for 24 h. In a comparison of the three mentioned chemical donors, it was found that the H_2_O_2_ solution was the best chemical donor for oxygen species molecules to enhance the seed germination percentage of rice. When comparing between the treatment times of 12 and 24 h to find an optimal time of treatment, we found that 24 h was the best amount of time for the treatment of rice seeds because the germination percentage showed both a rapid and high growth. However, the germination percentages among treatments and chemical donors were roughly the same at 24 h.

Figure 2 shows the germination percentages for rice seeds of the cultivar Sanpatong after treatment with H_2_O_2_, sodium nitroprusside, and sodium nitrite. After the concentration increased, H_2_O_2_ boosted the percentage of seed germination up to 99%, especially at 24 h of treatment and germination time (Figure 2A). The percentage of seed germination increased more rapidly when the treatment time of the rice seeds expanded from 12 to 24 h. However, there was no difference in the percentages of seed germination between treatments with sodium nitroprusside and sodium nitrite at either treatment time (Figure 2B,C). At 24 h after germination and a subsequent treatment with sodium nitroprusside, the germination percentage of the Sanpatong cultivar marginally decreased (Figure 2B). However, the germination percentages among the treatments and chemical donors were roughly the same at 72 h.

Furthermore, the germination percentage of the two rice cultivars, Pathum Tani and Sanpatong, can be raised and both of them respond to H_2_O_2_ better than sodium nitrite and sodium nitroprusside. The plant growth rate of the two rice cultivars, Pathum Tani and Sanpatong, after treatment with the three chemical donors on 21-day-old seedlings, did not differ significantly from the control, except for the Pathum Tani cultivar treated with sodium nitrite for 12 and 24 h. The comparison of the seedling lengths between the Pathum Tani seeds treated with sodium nitrite for 12 and 24 h demonstrated no difference among the treatment times and chemical concentrations (Figure 3).

### 2.2. Changes in Reducing Sugar and Total Soluble Protein during Rice Seed Germination

To investigate the effect of H_2_O_2_ on the chemical contents during germination, we measured the concentration of reducing the sugar (α-amylase activity indicator) and total soluble protein in germinated rice seeds at 24 and 48 h after treatment with H_2_O_2_ for 12 and 24 h. Figure 4A shows the enhancement effect of H_2_O_2_ on reducing the sugar concentration after a treatment with H_2_O_2_ for 12 and 24 h in the Pathum Tani cultivar. There was a trend toward reducing the sugar content after being significantly increased after the treatment with all concentrations of H_2_O_2_ solutions, especially 25 and 50 mM. The reducing sugar content was slightly reduced when the concentration of H_2_O_2_ was higher than 50 mM. These results correspond to the effect on the germination percentage seen in Figure 1A. The highest concentration of reducing sugar was 2516 µmol/g after treatment with 50 mM of H_2_O_2_ for 24 h. Moreover, the reducing sugar content was higher in seeds treated with H_2_O_2_ for 24 h than those treated for 12 h, at about twice as much. The reducing sugar from germinated seeds at 48 h was higher than that at 24 h.

Figure 5A shows the enhancement effect of H_2_O_2_ on reducing the sugar concentration after treatment with H_2_O_2_ for 12 and 24 h in the Sanpatong cultivar. Similar results have showed that, after treatment with H_2_O_2_ solutions, the reducing sugar content was also increased. The stimulating effect of H_2_O_2_ on the total soluble protein concentration in the Pathum Tani cultivar is depicted in Figure 4B. At 24 h of germination time, the total soluble protein when treated with H_2_O_2_ was significantly higher than in the control, at about 1.3–2.3 times for both 12 and 24 h of treatment time. At 48 h of germination time, the control treatment presented a comparable result to the H_2_O_2_ treatments at both 12 and 24 h of treatment time. The stimulating effect of H_2_O_2_ on the total soluble protein concentration in the Sanpatong cultivar is depicted in Figure 5B. After treatment with all H_2_O_2_ solution concentrations, the total soluble protein content slightly increased. The results of this study were in agreement with several reports [13,14] which showed that the reducing sugar and the total soluble protein contents rose during germination.

### 2.3. The Accumulation of ROS during Seed Germination on Two Rice Cultivars

The enhancement mechanism of reactive species for seed germination and the metabolism after treatment with H_2_O_2_, SNP, and sodium nitrite was evaluated. After the germination of the rice seeds for 3, 6, and 12 h without a seed coat, they were stained with NBT. Figure 6 shows that the plant radicle emerged from the seed coat and an embryo of the seed was stained with a dark blue color, which was localized in the seed axis. The dark blue staining rate displayed differently among germinated rice seeds treated with H_2_O_2_, SNP, and sodium nitrite (Figure 6A,B). In addition, the germinated rice seeds of the two rice cultivars treated with H_2_O_2_ presented a dark blue color on the radicle longer than those treated with sodium nitrite and SNP. When the two rice seed cultivars were compared, the germinated rice seeds of Pathum Thani showed stronger NBT chemical staining than the Sanpatong. It is noted that reactive oxygen and nitrogen species are continuously produced in the radicle of a plant during the germination process.

Moreover, the pH of the reactive chemical species may be another factor affecting the germination and metabolism of rice seeds. The pH measurement of H_2_O_2_, SNP, and sodium nitrite solutions was not changed among the different concentrations of the solution. The pH value of the H_2_O_2_ solutions was similar to that of distilled water, while SNP and sodium nitrite were slightly increased from 5.85 (DW) to 6.08 and 6.19, respectively. In other words, there was no significant difference between the control and the treatments (Figure 7).

## 3. Discussion

Seed germination is regulated by factors inside and outside of cells, such as genetics, phytohormones, seed coat, moisture, etc. It is clear that RONS play a key role in the seed germination stage by acting as a signaling molecule in the biological pathway of plants [15]. In this study, we investigated the effects of H_2_O_2_, SNP, and sodium nitrite on the rice seed germination of two types of rice: Pathum Tani (non-glutinous rice) and Sanpatong (glutinous rice). Glutinous rice differs from other types of rice in that it has very low amylose and high amylopectin contents [16]. Our results showed that H_2_O_2_ can enhance seed germination through changes in the chemical contents during the germination of seeds. Interestingly, rice seed treatment with H_2_O_2_ for the Pathum Tani cultivar demonstrated a faster germination than with sodium nitrite and SNP, approximately fourfold faster at 12 h of germination time. This evidence is supported by several reports which mentioned manifold metabolic processes during the seed germination of the plant [4,17]. Comparing the two rice cultivars after treatment with H_2_O_2_, Pathum Tani responded more quickly to H_2_O_2_ than Sanpatong at 12 h of germination time. This effect may be caused by the different kinds of polysachrides, amylose, and amylopectin in the two rice cultivars. Pathum Tani contains high amylose, which has a linear-chain structure, while Sanpatong contains high amylopectin, which has a branch-chain structure. In this case, the different polysaccharide structures are due to the action of the enzyme amylase, which breaks down starch during seed germination and seedling growth in rice [18]. The percentage of seed germination for Pathum Tani was significantly increased at a 25 mM concentration. However, a much higher concentration of H_2_O_2_ between 100 and 200 mM displayed a delayed seed germination process, suggesting that an optimal concentration of H_2_O_2_ should not be over 50 mM. It is noted that a higher dose of H_2_O_2_, similar to an exogenous reactive oxygen species, may negatively affect the seed vigor. Some research has revealed the effects of H_2_O_2_ and nitric oxide on seed germination and plant growth depending on the dose and the plant species [11,19]. In Sanpatong, the three RONS also improved the seed germination according to solution concentration and treatment time. Notably, Sanpatong strongly responded to high H_2_O_2_ concentrations along with the treatment time. Furthermore, the germination process involved phytohormones including abscisic acid (ABA) and gibberellic acids (GAs), activated by H_2_O_2_ and nitric oxide [4]. H_2_O_2_ regulated ABA catabolism by inhibiting activity; on the other hand, it activated GAs synthesis [20]. The role of ROS in the seed germination process is linked to dormancy release. During the imbibition stage, GAs are synthesized and triggers cellular metabolic processes to degrade the food storage of endosperm (monocot plant) or cotyledon (dicot plant) to aid cell growth in the germination stage [21].

In our study, imbibition with H_2_O_2_ promoted germination in rice seeds and was related to several chemical contents during germination, such as reducing sugar and total soluble protein. The quantity of reducing sugar, which is an indicator of the α-amylase activity, was significantly increased after the treatment with H_2_O_2_ solutions in accordance with the germination percentage. The results agreed with the findings for seed germination in upland rice seeds [13], mung bean seeds [22], and legume seeds [23], which showed a positive relationship between the germination and α-amylase activity. H_2_O_2_ imbibition may stimulate the activity of α-amylase through the gibberellins (GAs) hormone as follows: (1) the imbibition of water or H_2_O_2_ induces GAs synthesis within the embryo. (2) GAs stimulates the α-amylase production in the aleurone layer. (3) α-amylase hydrolyzes storage starch in the endosperm into sugar, which provides the energy for germination [24]. When the two rice cultivars were compared, the Pathum Tani had higher reducing sugar than the Sanpatong. This is most likely because Pathum Tani had a better germination percentage and higher metabolism than Sanpatong. The amount of total soluble protein in the Pathum Tani cultivar increased after the treatment with H_2_O_2_ at 24 h of germination time; meanwhile, there was no significant change across all treatments at 48 h of germination time. This may be caused by the synthesis of soluble protein in the early germination and it then remaining roughly constant during the following days. Some studies had a similar result in terms of showing that soluble protein was increasing during germination [14,25]. Their hypothesis was that the hydrolysis of glutelin produced free amino acids, which supported new soluble protein synthesis [25]. Previous studies have revealed that storage proteins are hydrolyzed by protease enzymes into soluble proteins, peptides, and free amino acids [26] and provide nutrients for seedlings growth [27,28]. Our results suggest that the chemical contents (reducing sugar and total soluble protein) increases during the germination of rice seeds. There are likewise several studies that have reported that the degradation of storage starch and protein in the endosperm provided energy and nutrients for seedling growth [13,22,25,29].

According to the obtained results, an accumulation of reactive oxygen species was found in germinated rice seed treated with H_2_O_2_, SNP, and sodium nitrite. Among the three chemical donors generating reactive species, we found that H_2_O_2_ well illustrated the role of ROS in the seed germination stage because the primary root emerging from the seed coat was stained dark blue with NBT dye. Previous studies have reported that germination and ROS accumulation appear to be linked, and seed germination success may be closely associated with intrinsic ROS contents and the activities of ROS-scavenging systems. Extrinsic ROS can induce early seed germination [11]. ROS accumulation in seeds is important for breaking the dormancy stage and stimulating seed germination [4]. In this experiment, H_2_O_2_ facilitated rapid seed germination, which triggers seed storage protein carbonylation (PCB) that induces the oxidative pentose phosphate pathway (oxPPP) for enhancing NADPH to thioredoxin (TRX) in favor of seed germination [4]. RONS functions as oxidative and nitrosative signaling molecules to induce seed germination within the oxidative aperture level [12]. In addition, RONS-mediated post-translational modifications (PTM), such as carbonylation, S-nitrosylation, and nitration, are attracting interest for their role in activating the seed germination process. Moreover, H_2_O_2_ is considered an important redox molecule generated from the physiological and biological metabolism of living cells, given its specific physical and chemical properties, including a remarkable stability within cells. H_2_O_2_ is the most stable ROS and easily migrates through cell membranes over long distances [30], and it has a half-life time of 10^–3^ s as well as rapidly reversible oxidation of the target proteins [8,31]. H_2_O_2_ can be transported by aquaporins localized in the cell membrane, not only causing long-distance oxidative damage [32,33], but also participating in cell signaling regulation [34] in various biological processes, namely, cell differentiation, development process, senescence, program cell death, cell wall formation, and stress responses in plants [35,36]. However, the ROS balance is necessary for cell adaptation and survival because an amount of ROS is necessary for diverse functions in the antioxidant system [30].

## 4. Materials and Methods

### 4.1. Plant Material and Treatment of Rice Seeds with Reactive Species Generated from Chemical Donors

Two rice seed cultivars, Pathum Tani (non-glutinous rice) and Sanpatong (glutinous rice), were used to evaluate the role of reactive species generated from three chemical donors (H_2_O_2_, sodium nitroprusside, and sodium nitrite) in seed germination and chemical contents. For seed treatment, a dilution of H_2_O_2_ solution (Merck, 2000 Kenilworth, NJ, USA) was used as the donor for H_2_O_2_, while sodium nitroprusside (SNP) and sodium nitrite (Sigma-Aldrich, St. Louis, MO, USA) were used as donors for nitric oxide (NO) molecules. All chemically reactive species donors were diluted with distilled water. For the H_2_O_2_ treatment on rice seeds, the seeds (150 seeds per treatment) were placed in glass bottles containing 50 mL of the H_2_O_2_ solutions of 25, 50, 100, and 200 mM, and then incubated at room temperature (dark condition) for 12 and 24 h. We dissolved SNP and sodium nitrite in distilled water to make the concentrations of 12.5, 25, 50, and 100 µM for the NO treatment. After placing the rice seeds in the glass bottles containing SNP and sodium nitrite solutions, the bottles were incubated at room temperature (dark condition) for 12 and 24 h. After the incubation period, the rice seeds were moved onto two layers of wet-cultivate paper (top of paper method) in a Petri dish (⌀ 9 cm, SPL Life Science, Gyeonggi-do, Korea), kept under dark conditions at room temperature, and then the germination was checked every day.

### 4.2. Analysis of the Percentage of Seed Germination and Growth

For the rice seed germination data analysis, we counted the number of germinating seeds every 3, 6, 12, and 24 h (Pathum Tani cultivar) and 12, 24, 48, and 72 h (Sanpatong cultivar) with the emergence of the embryo from the covering structure (rice husk) because the germination time of the two rice cultivars differed. After that, the seed germination data were calculated for the percentage of seed germination by following this formula:$$\mathrm{The}\;\mathrm{percentage}\;\mathrm{of}\;\mathrm{seed}\;\mathrm{germination}=\frac{\mathrm{Number}\;\mathrm{of}\;\mathrm{seed}\;\mathrm{germination}}{\mathrm{Total}\;\mathrm{number}\;\mathrm{of}\;\mathrm{seed}}\times 100$$

For the seedling plant growth rate measurement, after the rice seeds had a final germination period of five days, we planted them and grew them in a plastic chamber with soil, watering them daily. The seedling length of each treatment was then calculated by measuring the seedling plants when they were twenty-one days old. Each treatment consisted of fifty seedling plants, which were used to measure the plant growth.

### 4.3. Measurement of Reducing Sugar and Total Soluble Protein Contents

Rice seeds (Pathum Tani and Sanpatong) were treated with H_2_O_2_, SNP, and sodium nitrite solutions as described above. H_2_O_2_ treatment produced the highest percentage of germination among them. Therefore, we analyzed how the seed metabolism, including both reducing sugar and total soluble protein, was stimulated by the H_2_O_2_ solutions. Germinated rice seeds were harvested 24 and 48 h after the H_2_O_2_ treatment. Next, the seeds were dried at 60 °C for 24 h and were ground with a blender. A total of 6 mL of Tris-buffer solution (0.05 M Tris hydroxyl methionine, 0.02 M CaCl_2_, pH 7.4) was added to the ground powder for extracting and then the mixture was shaken for 1 h. The resulting homogenate was filtered and centrifuged at 2879× *g* for 5 min. The supernatant was transferred to a new tube and kept on ice for the analysis of the reducing sugar and total soluble protein. The reducing sugar contents were measured by the dinitrosalicylic acid (DNS) method [11]. A volume of 0.25 mL extract solution was mixed with 0.25 mL of Tris-buffer (ratio 1:1); thereafter, 0.5 mL of 1%w/v starch solution (starch in 0.02 M phosphate buffer, pH 7.0) was added. The mixture was then incubated at 37 °C for 1 h. After that, 0.5 mL of DNS (1%w/v of 3,5-dinitrosalicylic acid in 0.02 M NaOH and 30%w/v sodium potassium tartrate solution) was added to the mixture, heated at 100 °C for 5 min, and cooled at room temperature. After adding 5 mL of distilled water into the samples, the absorbance was measured at 540 nm using a spectrophotometer (Genesys 20, Thermoscientific, Waltham, MA, USA). The contents of reducing sugar were calculated from a calibration curve made by using maltose as a standard sugar. The Bradford protein assay was used to measure the concentration of total soluble protein with minor modifications of the Bradford method [37]. The extract solutions (50 µL) were mixed with 2.5 mL of a fivefold dilution of protein assay dye reagent concentrate (Bradford reagent, Bio-Rad, Hercules, CA, USA) and incubated at room temperature for 5 min. The absorbance was measured at 595 nm using a spectrophotometer. The amount of total soluble protein was calculated from a calibration curve by using bovine serum albumin (BSA) as a standard protein. For the Sanpatong cultivar, the amount of total soluble protein was measured at 280 nm by a spectrophotometer (Genesys 20, Thermoscientific, MA, USA). In total, 1 mL of the extract solutions were quantified by quartz cuvette. Then, the data were normalized for comparing between the two rice cultivars.

### 4.4. Measurement of ROS Storage in Seed by Using Nitro Blue Tetrazolium Staining and pH Solution Measurement

As for the measurement of reactive oxygen species, rice seeds were treated with the same conditions mentioned above. Then, the seeds germinated for 3, 6, and 12 h were stained with 6 mM of nitro blue tetrazolium (NBT) (Sigma-Aldrich, St. Louis, MO, USA), prepared in 10 mM of Tris-HCl buffer (pH 7.4) for 30 min at room temperature, and were kept in the dark. Next, the rice seeds were rinsed with distilled water once, and navy or dark spots in the embryo area of the seeds were noted. The pH of the solution was measured by using Laqua twin pH 22 (Horiba scientific, Kyoto, Japan).

### 4.5. Statistical Analysis

The data for the percentage of the rice seed germination, the seedling length, and the chemical contents were calculated as the mean and standard deviation (SD) for indicating the number of replicates (≥3). The statistical analysis was analyzed by using Student’s *t*-test to establish significance among the data points, and significant differences were based on the *p* < 0.05 or *p* < 0.01 (* *p* < 0.05 and ** *p* < 0.01).

## 5. Conclusions

The ROS (hydrogen peroxide, H_2_O_2_), and the RNS generated from SNP and sodium nitrite, enhanced the seed germination, growth, and the chemical contents of the rice seed in the cultivars Pathum Tani and Sanpatong. The RONS generated from the chemical donor treatment on these two cultivars of rice seeds increased the germination percentage, especially H_2_O_2_. The rice cultivar Pathum Tani, however, was the only one whose growth rate was accelerated by the sodium nitrite treatment. Comparing the three chemical donors, H_2_O_2_ showed the best efficacy of all the reactive species for activating early seed germination, followed by sodium nitrite and then SNP. The percentage of seed germination rose to 99% at 6 h of germination time after treatment with 25 mM of H_2_O_2_ concentration for 24 h. Additionally, the Pathum Tani seeds germinated faster than the Sanpatong seeds, while Sanpatong better tolerated H_2_O_2._ Moreover, the H_2_O_2_ application resulted in a significant increase in the α-amylase activity and total soluble proteins, and H_2_O_2 _presented a dark blue color on the emerging radicle longer than sodium nitrite and SNP. Our results suggest that the applied concentration of H_2_O_2_ is the best RONS treatment in our experiment to enhance the germination of the selected two rice cultivars.

## Figures and Tables

**Figure 1 plants-12-00765-f001:**
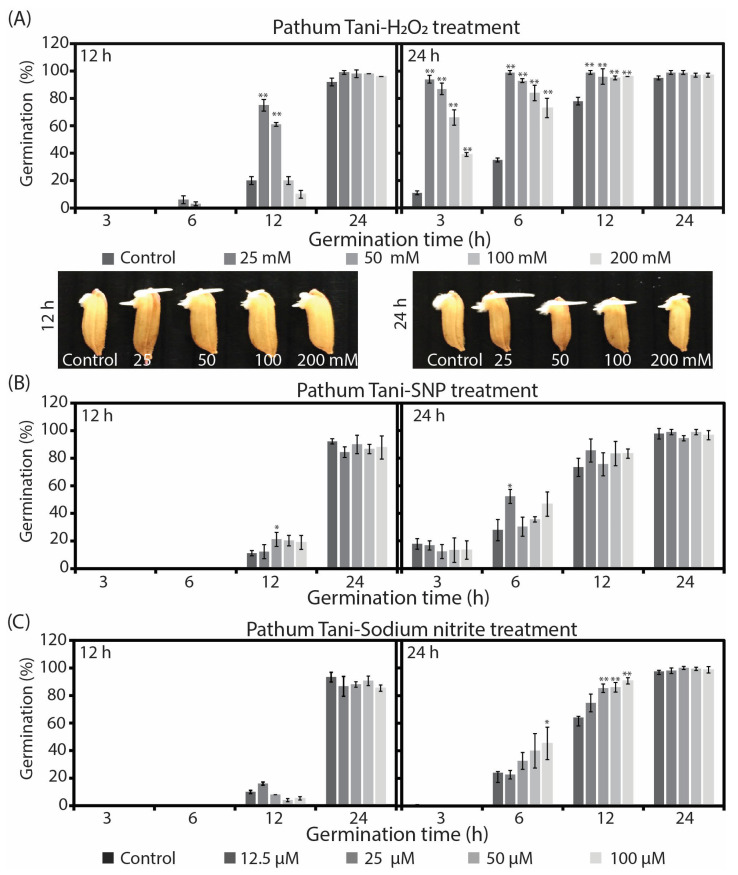
The effects of H_2_O_2_, SNP, and sodium nitrite on seed germination of Pathum Tani cultivar and seeds treated for 12 and 24 h. (**A**) The percentage of seed germination after treatment with H_2_O_2_ in concentrations 25, 50, 100, and 200 mM. (**B**) The percentage of seed germination after treatment with SNP in concentrations 12.5, 25, 50, and 100 µM. (**C**) The percentage of seed germination after treatment with sodium nitrite in concentrations 12.5, 25, 50, and 100 µM. (* *p* < 0.05 or ** *p* < 0.01).

**Figure 2 plants-12-00765-f002:**
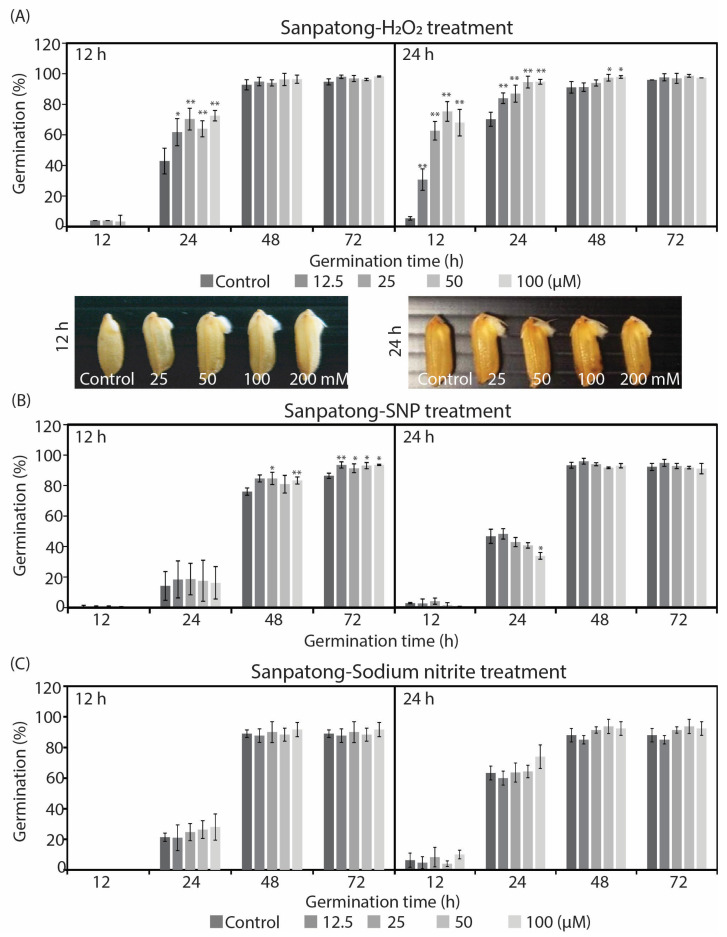
The effects of H_2_O_2_, SNP, and sodium nitrite on germination of Sanpatong cultivar and seeds treated for 12 and 24 h. (**A**) The percentage of seed germination after treatment with H_2_O_2_ in concentrations 25, 50, 100, and 200 mM. (**B**) The percentage of seed germination after treatment with SNP in concentrations 12.5, 25, 50, and 100 µM. (**C**) The percentage of seed germination after treatment with sodium nitrite in concentrations 12.5, 25, 50, and 100 µM. (* *p* < 0.05 or ** *p* < 0.01).

**Figure 3 plants-12-00765-f003:**
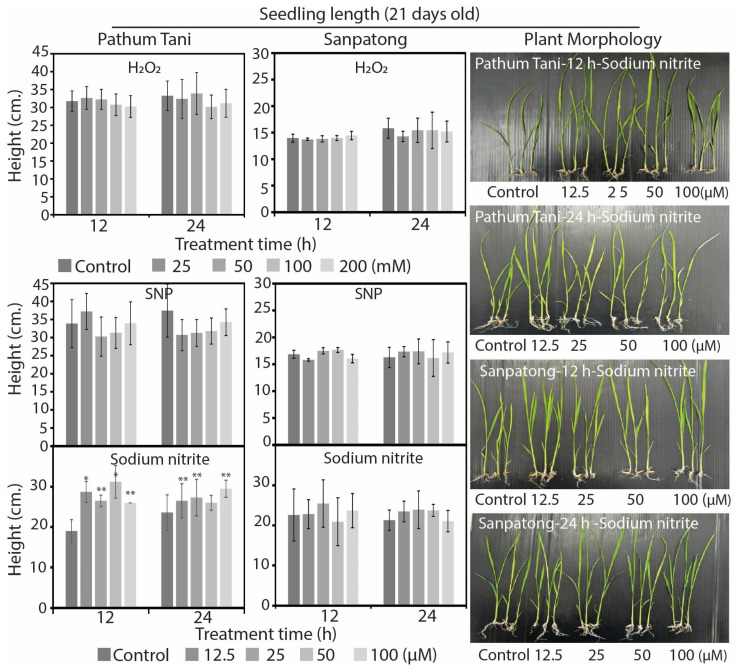
The effects of H_2_O_2_, SNP, and sodium nitrite solutions on the plant growth of two rice cultivars, Pathum Tani and Sanpatong, for 21-day-old plants at 12 and 24 h following treatment time (* *p* < 0.05 or ** *p* < 0.01).

**Figure 4 plants-12-00765-f004:**
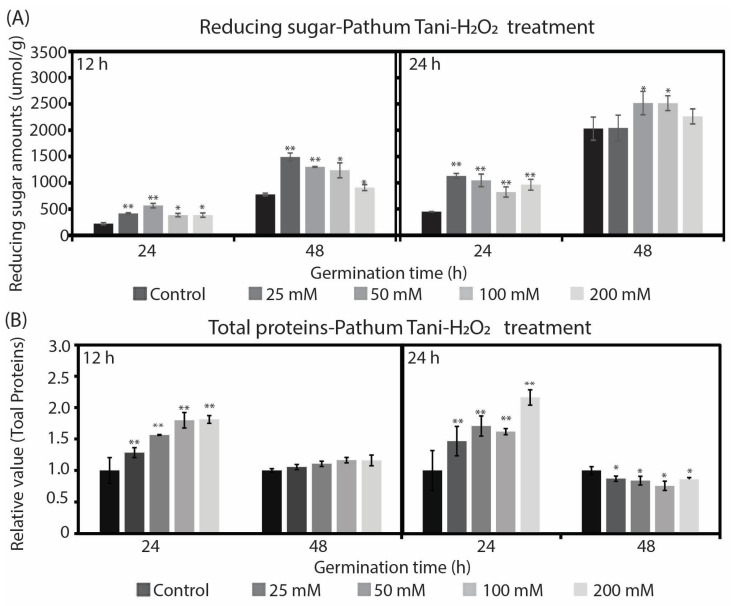
The effects of H_2_O_2_ solutions on the metabolism of germinated rice seed cultivar Pathum Tani after being treated with H_2_O_2_ for 12 and 24 h at 24 and 48 h post-germination. (**A**) The amount of reducing sugar in germinated rice seeds. (**B**) The amount of total soluble proteins in germinated rice seeds. (* *p* < 0.05 or ** *p* < 0.01).

**Figure 5 plants-12-00765-f005:**
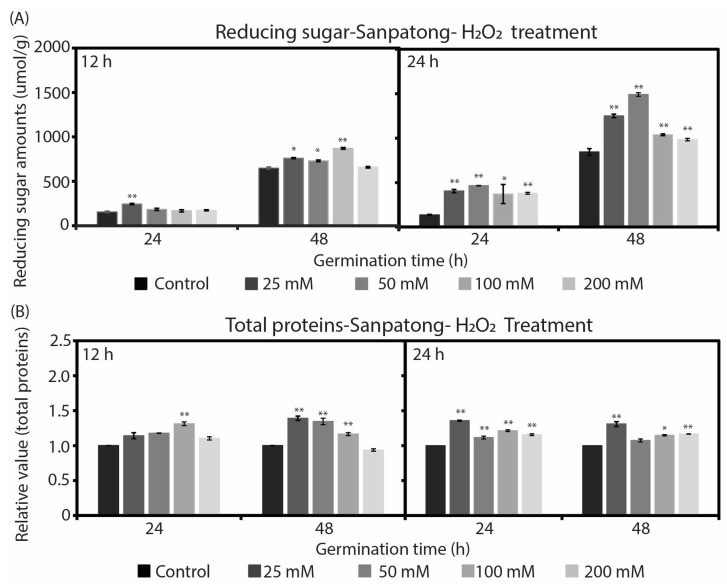
The effects of H_2_O_2_ solutions on the metabolism of germinated rice seed cultivar Sanpatong after being treated with H_2_O_2_ for 12 and 24 h at 24 and 48 h post-germination. (**A**) The amount of reducing sugar in germinated rice seeds. (**B**) The amount of total soluble proteins in germinated rice seeds. (* *p* < 0.05 or ** *p* < 0.01).

**Figure 6 plants-12-00765-f006:**
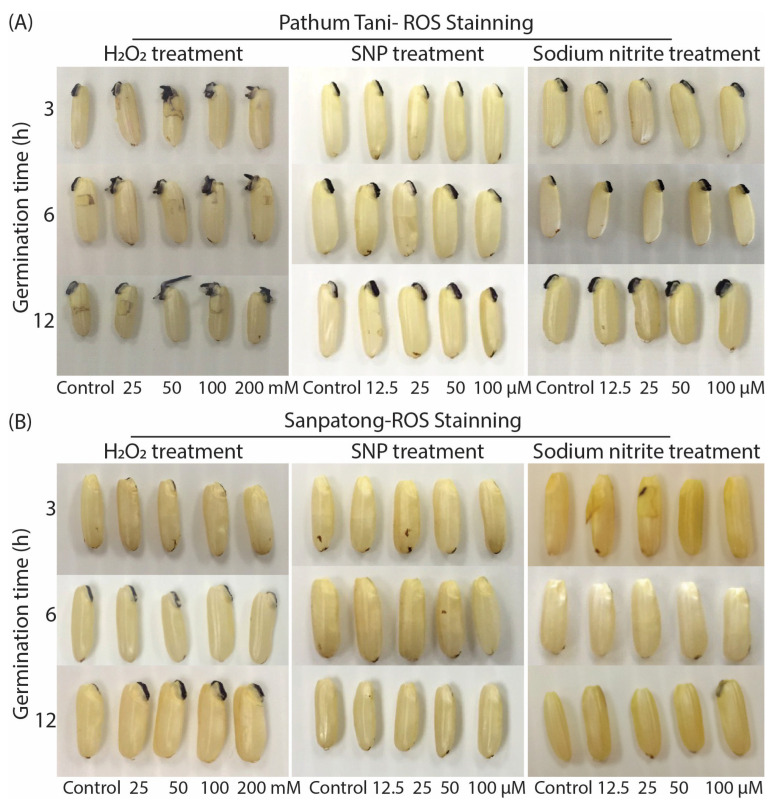
The amount of reactive oxygen species molecules accumulated in two rice seed cultivars Pathum Tani and Sanpatong. (**A**) Rice seeds (Pathum Tani) and (**B**) Rice seeds (Sanpatong) without seed coat were stained with 6 mM NBT dye for 30 min in the dark at room temperature and a dark blue color illustrated that the oxygen molecule reacts with NBT dye.

**Figure 7 plants-12-00765-f007:**
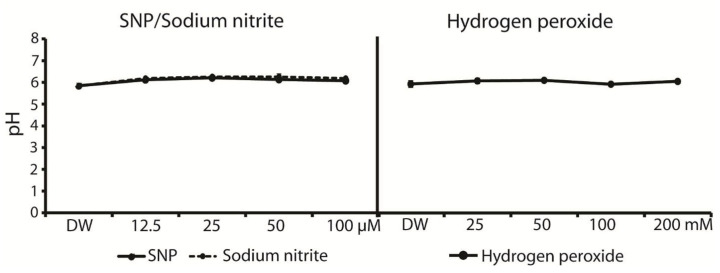
The pH of SNP, sodium nitrite, and H_2_O_2_ solutions in different concentrations and measured by using Laqua twin pH 22.

## Data Availability

Not applicable.

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
