# Peer review of "Different Effects of Reactive Species Generated from Chemical Donors on Seed Germination, Growth, and Chemical Contents of *Oryza sativa* L."

_plants, 2023, doi:10.3390/plants12040765_

Round 1

Reviewer 1 Report

Comments

In general, the submitted manuscript to Plants is set out to assess the effects of reactive species on growth and metabolism of Oryza sativa L.. I think that the paper has the publication potential, but it should be improved in various aspects, as mentioned in the following comments before final production:

Title should be modified and the use of abbrevations should be avoided.

Line 25: What was the rationale behind the activation of early seed germination by H2O2?

Line 31: „is” should be replaced by „had”

Line 71-73: Re-write

Line 76: remove „depending on the chemicals donors”. Please do follow the same practice for the subheadings of results section.

Line 133: metabolic term reflects the number/series of metabolites but the authors just measured few metabolites. On the basis of few metabolites, the authors can not mention the metabolism since the metabolism is a large biological phenomenon which needs several metabolites to be measured: at least which are commonaly involved in glycolysis, OPPP and calvin cycle.

Figures: significant letters should be mentioned for each figure.

Figure 3: Y-axis: Height*

Line 351-54: Re-write

Line 363-64: Re-write.

The authors are suggested to change the term „metabolism” until they measure several metabolites.

Extensive English editing is required for this manuscript to be considered for publication in Plants.

Author Response

We would like to thank the reviewers for very precious comments. The reviewer’s comments have been very helpful in the improvement of our manuscript. Here are our answers to reviewers’ questions and comments. Corrections and modifications in the manuscript text were indicated with an underline.

Reviewer #1:

In general, the submitted manuscript to Plants is set out to assess the effects of reactive species on growth and metabolism of Oryza sativa L. I think that the paper has the publication potential, but it should be improved in various aspects, as mentioned in the following comments before final production:

I. Title should be modified, and the use of abbreviations should be avoided.

Answer. As the reviewer suggested, we changed the research title from “Different effects of reactive species generated from chemical donors on seed germination and metabolism of Oryza sativa L.” to “Different effects of reactive species generated from chemical donors on seed germination, growth and chemical contents of Oryza sativa L.” We used abbreviations in the title because Oryza sativa L. is the scientific name for rice.

II. Line 25: What was the rationale behind the activation of early seed germination by H2O2?

Answer. The sentence in line 25 shift to line 26. We considered from the beginning of seed germination. In Pathum Tani, the percentage of rice seed germination significantly increased compared to control at 12 h of germination time after treatment with H2O2 solutions for 12 h. Among the three chemical donors, H2O2 promoted the quickest early seed germination of rice, followed by sodium nitrite and SNP, respectively. Pathum Tani and Sanpatong both showed similar results. For discussion section, we also mentioned H2O2 is play a role as an important redox molecule generated from the physiological and biological metabolism in plant cell. Additionally, H2O2 is the most stable ROS which easily migrate through cell membrane over long distance and has a half-life time of 10–3 s and rapidly reversible oxidation of target proteins. H2O2 can be transported by aquaporins localized in the cell membrane, not only causing long-distance oxidative damage, but also participating in cell signaling regulation in various biological processes. For above reasons exogenous H2O2 could be transport to inside seed cell through membrane with protein carrier (H2O2-transporting aquaporins). All reasons we added in the discussion part in line 280-289.

III. Line 31: „is” should be replaced by „had”

Answer. We replaced “is” by “had” in line 31.

IV. Line 71-73: Re-write

Answer. We rewrite the sentence and added a detail of research’s aim in line 70-76 (the sentence in line 71-73 shift to line 70-76). From the sentence “Therefore, in this research, focuses on reactive oxygen and reactive nitrogen species for determining the effectiveness of the types of reactive species as well as their doses as the activator and interaction with rice seeds.’’ to “Therefore, this research focuses on reactive oxygen and reactive nitrogen species to determine what types of reactive species as well as their doses serve as activators for seed germination and growth in non-glutinous (Pathum Tani) and glutinous (Sanpatong) rice. Additionally, we analyzed the chemical contents and the accumulation of ROS during seed germination.”

V. Line 76: remove „depending on the chemicals donors”. Please do follow the same practice for the subheadings of results section.

Answer. As the reviewer suggested, we removed “depending on the chemicals donors” and follow the same practice for the subheadings of results section in line 139 and 176.

VI. Line 133: metabolic term reflects the number/series of metabolites but the authors just measured few metabolites. On the basis of few metabolites, the authors can not mention the metabolism since the metabolism is a large biological phenomenon which needs several metabolites to be measured: at least which are commonly involved in glycolysis, OPPP and calvin cycle.

Answer. We agree with the reviewer’s opinion. As the reviewer suggested, we change subheading from “2.2 Metabolic changes during rice seed germination.” to “2.2 Changes in reducing sugar and total soluble protein during rice seed germination.” in line 139.

VII. Figures: significant letters should be mentioned for each figure.

Answer. The statistical analysis was analyzed by using a student’s t-test to establish significance among data points. Therefore, we indicated significant differences by “*” for p<0.05 and “**” for p<0.01 in figures.

VIII. Figure 3: Y-axis: Height*

Answer. We corrected “Heigh” to “Height” in Figure 3.

IX. Line 351-54: Re-write

Answer. As the reviewer suggested, we rewrite the sentence in line 351-354 shift to line 371-373 from “The role of ROS generated from hydrogen peroxide (H2O2), and RNS generated from SNP and sodium nitrite as to the enhancement of seed germination and metabolism of rice seeds in cultivar; Pathum Tani and Sanpatong.” to “The ROS generated from hydrogen peroxide (H2O2), and the RNS generated from SNP and sodium nitrite, enhanced seed germination, growth, and the chemical contents of rice seed in the cultivars Pathum Tani and Sanpatong.”

X. Line 363-64: Re-write.

Answer. We rewrite the sentence in line 363-364 shift to line 383-385 from “Our results suggest that H2O2 is the main role in enhancing germination and changing the metabolism of rice seeds.” to “Our results suggest that H2O2 is the best RONS for enhancing germination and changing the chemical contents of the two rice seed cultivars”

XI. The authors are suggested to change the term “metabolism” until they measure several metabolites.

Answer. We changed the term “metabolism” to “chemical contents” in title and in related text content.

XII. Extensive English editing is required for this manuscript to be considered for publication in Plants.

Answer. We submitted the revised manuscript for English editing to MDPI and already received the English Editing Certificate.

Reviewer 2 Report

The manuscript submitted by Chuesaard et al., is intended to present the results of their experiments on studying the effects of reactive species on rice germination parameters.

The manuscripts contain an adequate quantity of data and an acceptable structure from experimental design to the representation of the obtained results.  However, the investigated parameters are not adding too much to our current understanding about the effect of RONS on seed germination. The authors would have had the possibility to look at some more novel aspects of seed germination and associated enzymes activities in response to their priming trials to shed more lights to the actual physiology and biochemistry. Although quite a large volume of work has been done, the current manuscript is almost stays at an observational level as the presented discussion also is very week and cannot trigger new thought or give novel ideas or hypothesis about the actual events that are taking place as a result of conducted treatments. Speaking of the current manuscript and its content, here are some comments that authors must take into consideration before publishing this work.

The manuscript needs comprehensive English editing. There are several grammatical errors such as in ‘’ Our findings imply that H2O2 is a critical role in improving germination’’. These types of corrections can be done during the final English editing but in cases like: ‘’Therefore, in this research, focuses on reactive oxygen and reactive nitrogen species for determining the effectiveness of the types of reactive species as well as their doses as the activator and interaction with rice seeds.’’ the whole sentence is ambiguous and need a complete revision to be understandable. Therefor I do suggest revising the text thoroughly.

Line 82: Try to be consistent with the terminology. I suggest using gust the germination rate in in parentheses (%). Like higher, highest, lower, and lowest germination rate (%).  

Fig 1 and 2. Again try to follow a similar pattern. In fig 1, the cultivar name is missing where it is shown in Fig 2. In the Y-axis, just give the units. In Fig 3, change ‘’heigh’’ to ‘Height’, also ‘seedling length’ instead of ‘’plant growth’’.  Use the same shades of color in related charts like the Fig 1 and Fig 2. Figure captions must be improved. Shorten the text and keep the necessary information. For instance, there is nothing bout ‘’germinated seed morphology’’ in the Fig 1, especially if you have images from the seeds treated only with H2O2. The full set of similar images can be presented in the main text or come as supplementary materials. In Fig 4 give the total protein a unit like mg g-1 DW. And remove the ‘’Relative value’’.

The discussion is very inconsistent, and it contains similar information (duplicates) as given in the results section. Try to discuss your underrating from the obtained results with the other reports of similar experiments. Wherever you bring definitive statements like: ‘’Furthermore, the germination process involved phytohormone between abscisic acid (ABA) and gibberellic acid (GA) which is activated from H2O2 and nitric oxide’’ or in line 207-208 ’’This evidence is supported by several reports…’’ give reference/s.

Line 314: why liquid N was used for the samples that were already heat dried??? Rather delete it part if you cannot justify it.

Line 319: Give a reference for DNS methods and for using the selected concentrations of the applied RONSs.

All in all, the text be significantly revised to reflect the true content of your work in a logical and understandable manner for potential readers.

Author Response

We would like to thank the reviewers for very precious comments. The reviewer’s comments have been very helpful in the improvement of our manuscript. Here are our answers to reviewers’ questions and comments. Corrections and modifications in the manuscript text were indicated with an underline.

Reviewer #2:

I. The manuscript needs comprehensive English editing. There are several grammatical errors such as in ‘’ Our findings imply that H2O2is a critical role in improving germination’’. These types of corrections can be done during the final English editing but in cases like: ‘’Therefore, in this research, focuses on reactive oxygen and reactive nitrogen species for determining the effectiveness of the types of reactive species as well as their doses as the activator and interaction with rice seeds.’’ the whole sentence is ambiguous and need a complete revision to be understandable. Therefore, I do suggest revising the text thoroughly.

Answer. As the reviewer suggested, we sent the revised manuscript to MDPI for English editing and we rewrite the sentence “Therefore, in this research, focuses on reactive oxygen and reactive nitrogen species for determining the effectiveness of the types of reactive species as well as their doses as the activator and interaction with rice seeds.’’ to “Therefore, this research focuses on reactive oxygen and reactive nitrogen species to determine what types of reactive species as well as their doses serve as activators for seed germination and growth in non-glutinous (Pathum Tani) and glutinous (Sanpatong) rice. Additionally, we analyzed the chemical contents and the accumulation of ROS during seed germination.”

II. Line 82: Try to be consistent with the terminology. I suggest using gust the germination rate in in parentheses (%). Like higher, highest, lower, and lowest germination rate (%).  

Answer. As the reviewer suggested, we used the germination rate in parentheses (%) in line 84-86 of the sentence “Rice seeds being treated with H2O2 for 24 h accelerated the speed of germination rate at 3 h of germination time (94%) and the highest (99%) at 6 h of germination time.”

III. Fig 1 and 2. Again try to follow a similar pattern. In fig 1, the cultivar name is missing where it is shown in Fig 2. In the Y-axis, just give the units. In Fig 3, change ‘’heigh’’ to ‘Height’, also ‘seedling length’ instead of ‘’plant growth’’.  Use the same shades of color in related charts like the Fig 1 and Fig 2. Figure captions must be improved. Shorten the text and keep the necessary information. For instance, there is nothing bout ‘’germinated seed morphology’’ in the Fig 1, especially if you have images from the seeds treated only with H2O2. The full set of similar images can be presented in the main text or come as supplementary materials. In Fig 4 give the total protein a unit like mg g-1 DW. And remove the ‘’Relative value’’.

Answer. As the reviewer suggested, we edited figure 1 and 2 for similar pattern of the bar graph and improved the figure captions by deleted “germinated seed morphology”. Also, we changed ‘’heigh’’ to ‘Height’ already in the figure 3. In case of Figure 4 we cannot change the “Relative value’’ of Y-axis because we analyzed the total soluble protein of the two rice cultivars by difference method and different times. Bradford method was used for analysis of the total soluble proteins in Pathum Tani cultivar while measurement of absorbance at 280 nm was used for Sanpatong cultivar. Reanalysis may display different experimental results due to some factors such as seed age and storage conditions of rice seeds, effect of temperature for germination study, etc. Therefore, data from the two methods were normalized for comparing between the two rice cultivars in “relative value” for observation of total soluble protein trend. We also added measurement of absorbance at 280 nm for Sanpatong cultivar in line 351-354. 

IV. The discussion is very inconsistent, and it contains similar information (duplicates) as given in the results section. Try to discuss your underrating from the obtained results with the other reports of similar experiments. Wherever you bring definitive statements like: ‘’Furthermore, the germination process involved phytohormone between abscisic acid (ABA) and gibberellic acid (GA) which is activated from H2O2and nitric oxide’’ or in line 207-208’’ This evidence is supported by several reports…’’ give reference/s.

Answer. We tried to discuss our results with other reports of similar experiment and added discussion in line 209-223 and line 283-284. We also gave reference in line 214-215 “This evidence is supported by several reports which mentioned manifold metabolic processes during seed germination of the plant [4,17].” and line 231-233 “Furthermore, the germination process involved phytohormone between abscisic acid (ABA) and gibberellic acid (GA) which is activated from H2O2 and nitric oxide [4].”

V. Line 314: why liquid N was used for the samples that were already heat dried??? Rather delete it part if you cannot justify it.

Answer. As the reviewer suggested, we deleted “in liquid nitrogen” and modified the sentence to “Next, seeds were dried at 60oC for 24 h and were ground with a blender.” The sentence in line 314 shift to line 331.

VI. Line 319: Give a reference for DNS methods and for using the selected concentrations of the applied RONSs.

Answer. We gave reference for DNS method in the sentence of line 336 “Reducing sugar contents were measured by the dinitrosalicylic acid (DNS) method [11].” (The sentence in line 319 shift to line 336.) For selected the concentrations of RONSs, we chose the concentration of chemical donors from our previous laboratory experiments which was already found the optimum concentration range for testing in both dicotyledon (carrot, chili, coriander) and monocotyledon (rice, grass) plants.

Round 2

Reviewer 2 Report

The manuscript quality and organization has been changed and improved significantly. Therefore, I suggest the submitted manuscript by Chuesaard et al., to be accepted for publication in Plants.

Line 380: Change: ‘’Pathum Tani more speedily germinated seed than Sanpatong, while Sanpatong better …’’ to ’Pathum Tani seeds germinated faster than the Sanpatong seeds, while Sanpatong better…’

Line 28 and 381: Change‘’…produced…’’ to ‘…H2O2 application resulted in… ‘. However, Try NOT to use the same sentences in both abstract and conclusions.

Line 383-384: Modify to: ‘Our results suggest that the applied concentration of H2O2 is the best RONS treatment in our experiment to enhance the germination of the selected two rice cultivars.’

Author Response

We thank the reviewer very much for the comment. We modified the sentence as the reviewer suggested in lines 380, 381, and 383-384 and indicated with an underline.